# Pulsed-Xenon Ultraviolet Light Highly Inactivates Human Coronaviruses on Solid Surfaces, Particularly SARS-CoV-2

**DOI:** 10.3390/ijerph192113780

**Published:** 2022-10-23

**Authors:** Melissa Bello-Perez, Iris Esparza, Arancha De la Encina, Teresa Bartolome, Teresa Molina, Elena Sanjuan, Alberto Falco, Luis Enjuanes, Isabel Sola, Fernando Usera

**Affiliations:** 1Department of Molecular and Cell Biology, National Center of Biotechnology (CNB-CSIC), Campus Universidad Autónoma de Madrid, Darwin 3, 28049 Madrid, Spain; 2Biosafety Service, National Center of Biotechnology (CNB-CSIC), Campus Universidad Autónoma de Madrid, Darwin 3, 28049 Madrid, Spain; 3CandelTEC S.L. Pol. Industrial L’Horta Vella, 8, 6, 46117 Betera, Spain; 4Institute of Research, Development and Innovation in Healthcare Biotechnology in Elche (IDiBE), Miguel Hernández University (UMH), 03202 Elche, Spain

**Keywords:** coronavirus, HCoV-229E, MERS-CoV, SARS-CoV-2, pulsed-xenon ultraviolet, virus inactivation

## Abstract

In the context of ongoing and future pandemics, non-pharmaceutical interventions are critical in reducing viral infections and the emergence of new antigenic variants while the population reaches immunity to limit viral transmission. This study provides information on efficient and fast methods of disinfecting surfaces contaminated with different human coronaviruses (CoVs) in healthcare settings. The ability to disinfect three different human coronaviruses (HCoV-229E, MERS-CoV, and SARS-CoV-2) on dried surfaces with light was determined for a fully characterized pulsed-xenon ultraviolet (PX-UV) source. Thereafter, the effectiveness of this treatment to inactivate SARS-CoV-2 was compared to that of conventional low-pressure mercury UVC lamps by using equivalent irradiances of UVC wavelengths. Under the experimental conditions of this research, PX-UV light completely inactivated the CoVs tested on solid surfaces since the infectivity of the three CoVs was reduced up to 4 orders of magnitude by PX-UV irradiation, with a cumulated dose of as much as 21.162 mJ/cm^2^ when considering all UV wavelengths (5.402 mJ/cm^2^ of just UVC light). Furthermore, continuous irradiation with UVC light was less efficient in inactivating SARS-CoV-2 than treatment with PX-UV light. Therefore, PX-UV light postulates as a promising decontamination measure to tackle the propagation of future outbreaks of CoVs.

## 1. Introduction

The main route of human-to-human transmission of respiratory coronaviruses (CoVs), as well as some other infectious agents, is through respiratory droplets and aerosols generated by infected individuals. Virus transmission also often occurs through direct contact with contaminated surfaces, especially in crowded facilities and hospital environments where the loads of free virions may reach critical levels at certain time points [1,2,3]. In addition, infectious agents can remain infectious on inanimate surfaces at room temperature for considerable periods—in particular, for up to 9 days for human CoVs, including the severe acute respiratory syndrome CoV 2 (SARS-CoV-2), the causative agent of the coronavirus disease 2019 (COVID-19) [1,2,3]. Altogether, these events call for urgently reducing the dissemination of pathogens through rapid disinfection methods, such as light, and particularly ultraviolet (UV) irradiation [4,5,6,7].

The spectral range of UV light, with wavelengths between ~10 and 400 nm, corresponds to the electromagnetic radiation frame with the most energized photons that interact with matter by mostly electron excitation, rather than ionization [8]. Based upon their effects on (especially organic) molecules, UV light is generally divided into four bands: near (400–300 nm), middle (300–200 nm), far (200–100 nm), and extreme (below 100 nm) [6,8]. The region of shorter wavelengths, ranging up to 200 nm, is commonly known as vacuum UV (VUV) due to the high absorption of its rays by oxygen, which strongly limits their penetration in air and induces the generation of harmful levels of ozone. This prevents its implementation for common uses despite being the most energetic photons within the UV spectrum [6,8]. Research on the germicidal properties of UV light has, therefore, mainly focused on the bands of less energetic wavelengths: UVC, from 200 to 280 nm; UVB, 280–315 nm; and UVA, 315–400 nm (in descending order of energy) [5,6,8].

Among them, UVC is the principal UV germicidal irradiation (UVGI) because of its direct antimicrobial properties. Its underlying mechanism of action is based on its efficient absorption by unsaturated organic compounds and the consequent disarrangement of their conjugated bonds that particularly and critically affect nucleic acids [9]. The use of UVC is, therefore, dominant in commercial systems for disinfection purposes [5,6,8]. UVB also presents such direct antimicrobial properties, although to a much lesser extent. However, both UVB and UVA can indirectly inactivate microbes via oxidative damage by inducing the generation of reactive oxygen species (ROS) through the excitement of endogenous chromophores. In fact, sunlight, the content of which is neglectable in UVC at the Earth’s surface as it is blocked by the atmosphere, has been shown to inactivate a broad range of microbes, including viruses [5,6,8]. In this sense, it is also noteworthy that UVB and UVA also present greater matter-penetration power than UVC; thus, it is potentially useful for applications requiring higher disinfection outputs in liquids and solid surfaces [8].

New UV light lamps are sophisticated robots implementing pulsed light in the whole spectrum and offering superior disinfectant properties for conventional lamps emitting restricted UV wavelengths. Pulsed-light treatment consists of short, high-frequency and high-intensity pulses of broad-spectrum light that normally range from approximately 200 to 1100 nm wavelengths, thus encompassing UV, visible, and infrared bands. This type of light dosage is usually supplied by pulsed-xenon lamps and termed pulsed-xenon UV (PX-UV) light. In addition to requiring lower energy consumption than continuous irradiation, while providing higher energy intensity and a wider range of the light spectrum, pulsed-light treatments have commonly been reported to provide superior (i.e., faster, stronger, and broader) germicidal efficiencies [4,6].

Limited information is available on the sensitivity of CoVs to PX-UV light [4,5,6,10], as only a few studies have addressed this topic [11,12,13,14]. In addition, to our knowledge, only one study has compared the inactivation of CoVs (three variants of SARS-CoV-2) by pulsed and continuous light, using a deep-UV light-emitting diode (DUV-LED) source of irradiating UVC light (250–300 nm), and reported no significant differences between them [15]. In this manuscript, the effectiveness of (broad-spectrum) PX-UV light in the inactivation of three human CoVs (i.e., HCoV-229E, MERS-CoV, and SARS-CoV-2), deposited as dry particles on a plastic surface, was analyzed to evaluate its potential to disinfect rooms in comparison to UVC from conventional low-pressure mercury lamps. Additionally, this work fully characterizes the emission source (including the emission spectrum, emission field, dose, and irradiance), thus allowing for the reproduction of the observed effects with any other device. The experimental setup also uses distances beyond those typically used in laboratory-scale experiments in order to approach sanitization conditions in real-life settings.

## 2. Materials and Methods

### 2.1. Virus Strains

SARS-CoV-2 was isolated from the nasopharyngeal swab of a 69-year-old male COVID-19 patient from Madrid (Spain) collected by a healthcare worker in February 2020. The nasopharyngeal swab was inserted in one nostril until reaching the back of the cavity and rotated before removal. The sample was taken 5 days postdiagnosis with an ultra-thin applicator swab with a flocked nylon fiber tip in 1 ml liquid Amies medium (Eswab Collection system, Copan, Italy, catalog No. 483C). The patient signed an informed consent form, and his confidential data were preserved. The procedure was approved by the Ethics Committee of Hospital 12 de Octubre, Madrid, Spain (code 20/101).

Recombinant MERS-CoV was rescued from an infectious cDNA generated using a bacterial artificial chromosome (BAC)-based reverse genetic system. HCoV-229E was kindly provided by Volker Thiel (Institute of Virology and Immunology, University of Bern, Swiss). Virus stocks were prepared as previously described [16]. All the experiments with SARS-CoV-2 and MERS-CoV were performed in biosafety level 3 (BSL-3) facilities at CNB-CSIC according to the guidelines of the institution.

### 2.2. Cell Lines

Human liver-derived Huh-7 cells susceptible to infection by HCoV-229E and MERS-CoV were kindly provided by Dr. L Carrasco (Centro de Biología Molecular Severo Ochoa, Madrid, Spain). Vero E6 cell lines, isolated from kidney epithelial cells extracted from an African green monkey, were kindly provided by Dr. E Snjider (Leiden University Medical Center, Netherlands) and were used to grow SARS-CoV-2.

All cell lines were cultured in Dulbecco’s modified Eagle medium (DMEM) supplemented with 2 mM l-glutamine (Sigma-Aldrich, St. Louis, MI, USA), 1% non-essential amino acids (Sigma-Aldrich), 10% fetal bovine serum (FBS; BioWhittaker, Inc., Walkersville, MD, USA), and antibiotics (100 units/mL penicillin and 100 μg/mL streptomycin) (Thermo Fisher, Waltham, MA, USA). They were used for virus titration using a plaque assay method. Briefly, cells infected with serial dilutions of virus inoculum were overlaid with DMEM containing 0.6% low-melting agarose or 0.7% melting agar, for SARS-CoV-2 and HCoV-229E or MERS-CoV, respectively, and 2% FBS. At 96 h post-infection, cells were fixed with 10% formaldehyde and stained with 0.1% crystal violet to visualize and count the viral plaques.

### 2.3. Inactivation of Coronavirus by UV-Irradiation

To evaluate the susceptibility of human CoVs to PX-UV or UVC light, 90 mm-diameter culture plates (Nunclon Delta Surface, Thermo Fisher) were irradiated with either a TUV PI-L 36W, UV-C (conventional UVC low-pressure mercury lamp, Philips, Netherlands), or Light Strike™ Germ-Zapping™ Robot (PX-UV lamp, Xenex Disinfection Services, San Antonio, TX, USA), respectively. The plates were placed at a distance of 1.5 or 2 m, facing the corresponding devices, at a 45° angle from the horizontal plane, as shown in the representative schemes from Figure 1A. The conventional UVC low-pressure mercury lamp was placed at 2 m of the experimental sample, and the PX-UV lamp was placed at 1.5 m because the doses of UVC light reported by the manufacturers for each device were approximately equivalent at those distances (Appendix A). As it is described later in this section, these data were further confirmed by measuring the irradiance received from experimental samples under this setup.

For the preparation of the experimental samples, a descriptive schematic diagram of this experimental setup is shown in Figure 1B. Briefly, a 320 μL drop of DMEM, containing 10^5^ plaque-forming units (PFU) of HCoV-229E, MERS-CoV, or SARS-CoV-2, was deposited on the culture plates and spread using an inoculation loop. Then, the plates were left to air dry for 45 min, since it had been assessed, for HCoV-229E as the virus model, that the initial viral titer was reduced by only one log unit in this period. Dried samples were irradiated with PX-UV light (200–600 nm, frequency between pulses of 67.35 Hz) or UVC light (254 nm) and collected immediately after 0, 0.5, 1, 2, 3, 4, and 5 min by washing the plate surface with 500 μL of phosphate-buffered saline (PBS) solution. For each time point, corresponding mock-irradiated virus samples were included in a closed plate and were processed in the same way as the experimental ones. Virus inactivation was determined as the reduction in the number of PFU after treatment in comparison to their corresponding mock-irradiated controls and expressed as log reduction units, calculated as follows: log_10_ (A/B), where A is the viral titer of a mock-irradiated sample, and B is the viral titer after light irradiation. All experiments were performed three times, each in triplicate, and the data are presented as the mean and s.d.

### 2.4. Quantification of the Irradiance Supplied to the Experimental Samples

The StellarNet Blue-Wave Spectroradiometer (200–600 nm) was used to determine the direct radiation on the virus-inoculated surfaces with the Light Strike™ Germ-Zapping™ Robot or the TUV PI-L 36W, UV-C lamps (Appendix A). This detector was anchored to a support that allowed its adequate location with respect to the source of light. With this setup, measurements were carried out at an inclination of 45° with respect to the horizontal plane and at a distance of 1.5 or 2 m from the Light Strike™ Germ-Zapping™ Robot or the TUV PI-L 36W, UV-C lamps, respectively (Figure 1A). Doses corresponding to each segment of the UV spectrum (UVA, UVB, and UVC) were calculated considering the time of exposure (Appendix A). For that, each irradiance value was multiplied by the time of exposure in seconds and a correction factor, provided by the manufacturer, that considers the fluctuation of the lamps during the initial period was applied.

### 2.5. Statistical Analysis

Student’s t-tests were used to analyze differences in mean values between the UV-treated and control groups at each time point. All results were expressed as means ± standard deviations. Differences between the treated and control groups were considered statistically significant when the *p*-value was less than 0.05 (*p* < 0.05, *; *p* < 0.01, **; *p* < 0.001, ***).

## 3. Results

### 3.1. Inactivation of CoVs on Contaminated Surfaces Using PX-UV Irradiation

HCoV-229E, from the CoV genus Alpha, and MERS-CoV and SARS-CoV-2, both belonging to CoV genus Beta, have been used as models to assess the efficacy of PX-UV light on the inactivation of CoVs on solid surfaces. The viral loads, before and after the irradiation, for up to 5 min of the virus inoculums, at a distance of 1.5 m (total UV doses from 2.810 to 26.381 mJ/cm^2^; UVC dose from 0.717 to 6.734 mJ/cm^2^), are shown in Table 1.

When PX-UV light was irradiated, viral titers of the three CoVs were reduced, even for short exposure times. The estimated time for 99.9% inactivation (3 log_10_ unit reduction) was 1 min (total UV dose: 5.448 mJ/cm^2^; UVC dose:1.391 mJ/cm^2^) for MERS-CoV and 2 min (total UV dose: 10.685 mJ/cm^2^; UVC dose: 2.728 mJ/cm^2^) for HCoV-229E and SARS-CoV-2. The amount of infectious virus was below the detection limit of the plaque assay (20 pfu/mL) after 2 min exposures for MERS-CoV, 3 min for SARS-CoV-2, and 4 min for HCoV-229E. Therefore, the three human CoVs were susceptible to inactivation with PX-UV radiation, although to different extents, with MERS-CoV being the most susceptible and HCoV-229E the most resistant.

### 3.2. Comparison between PX-UV or Conventional UVC Radiation in the Inactivation of SARS-CoV-2

SARS-CoV-2 has been used as a model to compare the efficacy of PX-UV light and conventional UVC low-pressure mercury lamps on the inactivation of CoVs on solid surfaces. According to the results presented here, the irradiation of samples contaminated with SARS-CoV-2 using a conventional UVC lamp at 2 m showed a 1.9 log reduction in comparison with the >3.8 log reduction obtained by irradiating for 5 min with the PX-UV light using an equivalent dose (Figure 2). This suggests that PX-UV irradiation is a more effective method of disinfecting contaminated surfaces.

## 4. Discussion

This manuscript demonstrates that PX-UV light is a rapid and effective method of disinfecting surfaces contaminated with human CoVs, even more than low-pressure mercury UVC lamps, to control viral propagation in healthcare settings or poorly-ventilated spaces. The infectivity of three different CoVs (HCoV-229E, SARS-CoV-2, and MERS-CoV) was reduced up to 3.8 log_10_ units by PX-UV irradiation applied for 4 min at a distance of 1.5 m from the device (total UV dose: 21.162 mJ/cm^2^; UVC dose: 5.402 mJ/cm^2^). These results are consistent with those observed in other reports, demonstrating that a pulse-on time of 5 min of UVC at a 1 m distance was sufficient to achieve a 3 log_10_ reduction in SARS-CoV-2 on plastic surfaces [14]. Results on the inactivation of HCoV-229E and MERS-CoV virions on dry surfaces have been performed in this work for the first time.

The effectiveness of PX-UV in inactivating CoV infectivity was expected because viruses with large, single-stranded RNA (ssRNA) genomes are the most susceptible to UVC radiation, and CoVs are the largest ssRNA RNA viruses known to date [17]. Differences in susceptibility to PX-UV radiation between CoVs were not expected in terms of RNA inactivation. All coronaviruses have similar RNA genomes, with a length of 27–32 kb [18]; consequently, they should exhibit a similar level of UVC absorption. Furthermore, the negative correlation between G + C content and the inactivation of ssRNA by UVC [18] does not explain those differences, as SARS-CoV-2 and HCoV-229E have identical G+C content (38%) [19,20]. However, the effects of UVC light on biological components also involve proteins [21]. CoV entry into cells depends on the specific cleavage of S glycoprotein to enhance the fusion of viral and host cell membranes. CoV S protein contains at least two cleavage sites, S1/S2 and S2′ [22]. Therefore, the cross-linking of S cleavage products produced by UVC light could affect the viral entry of CoVs and reduce infectivity. This effect could be less efficient in the case of HCoV-229E because S1/S2 cleavage does not seem to be essential for fusion activation, as it is highly dependent on proteolytic processing at S2′ [23]. This difference could explain the slight resistance of HCoV-229E to UVC irradiation in comparison to MERS-CoV and SARS-CoV-2, although alternative inactivation mechanisms cannot be excluded.

Additionally, a recent paper describes the germicidal effects of monochromatic UVA (λ = 366 nm) against CoVs in aerosols after 2 min of irradiation using doses of 2.34 mJ/cm^2^ [17]. Despite those results being in aerosols instead of surfaces, that work provides evidence of the inactivation of SARS-CoV-2 using UVA. Therefore, it is not possible to discard that part of the effect observed in our study is due to UVA light, since our system also emits in those conditions (Appendix A). In terms of UVB light, a recent study demonstrated that 90% of infectious SARS-CoV-2 dried on surfaces was inactivated every 14.3 min when exposed to UVB light (0.16 mW/cm^2^) [17]. Likewise, it has been described that a continuous UVB flux of 90 μW/cm^2^ resulted in an exponential decline in infectivity with the time of other viruses, such as virulent Newcastle disease viruses (NDVs) and the highly pathogenic avian influenza virus (HPAI), after more than one hour of exposure. These studies are difficult to compare with the results presented in this manuscript because they use continuous instead of pulsatile exposures, longer exposure times, and lower doses than those used here [17]. However, they show antiviral effects of UVB light that could explain part of the results presented in this study. In addition, since studies performed using PX-UVC light against CoVs do not describe the dose that each sample received, it is not possible to compare whether PX-UV is more effective than PX-UVC irradiation [17].

This study may have a significant impact on the disinfection of sanitary spaces in constant contact with the virus during pandemics, such as ambulances, hospital rooms, operating rooms, emergency rooms, or even the changing rooms where EPIs are discarded. However, UV irradiation must be used in empty spaces, out of contact with humans, because UV-B and UV-A are generally considered to be carcinogenic and UV-C light causes erythema and triggers photokeratitis [24]. PX-UV irradiation for 4 min at a 1.5 m distance (UV dose 21.162 mJ/cm^2^; UVC dose 5.402 mJ/cm^2^) seems to be enough, and more effective than irradiation with conventional UV lights, to effectively disinfect surfaces in closed spaces during pandemics. As this study fully describes the specific characteristics of the emission (emission spectrum, emission field, dose, and irradiance administered to the samples), they could be reproduced by any source of PX-UV light and to be included as a strategy in the infection control plans at hospitals. Unfortunately, UV disinfection has limitations under non-optimal conditions, such as the presence of dust or soil and the irregularities at microscopic levels in the surfaces, which create shadows [25]; however, the increased effectiveness of the PX-UV light—together with the implementation of robotic displacement technologies and new designs, including multiple lamps and/or reflective walls—could help overcome these hurdles.

## 5. Conclusions

This work concludes that human coronaviruses (HCoV-229E, SARS-CoV-2, and MERS-CoV) are susceptible to PX-UV inactivation, although differences in disinfection efficacy were observed depending on the virus and time exposure, HCoV-229E being the most resistant. A cumulative dose of as much as 21.162 mJ/cm^2^, considering all the UV wavelengths, is enough to reduce 4 orders of viral titers. The inactivation of SAR-CoV-2-contaminated surfaces is more efficient with PX-UV than with conventional low-pressure mercury lamps (>3.8 vs. 1.9 log reduction). In addition, this work comprehensively characterizes the irradiation method, which allows for the reproduction of the same effect for surface disinfection using different sources, provided that the emission of PX-UV is as described here.

## Figures and Tables

**Figure 1 ijerph-19-13780-f001:**
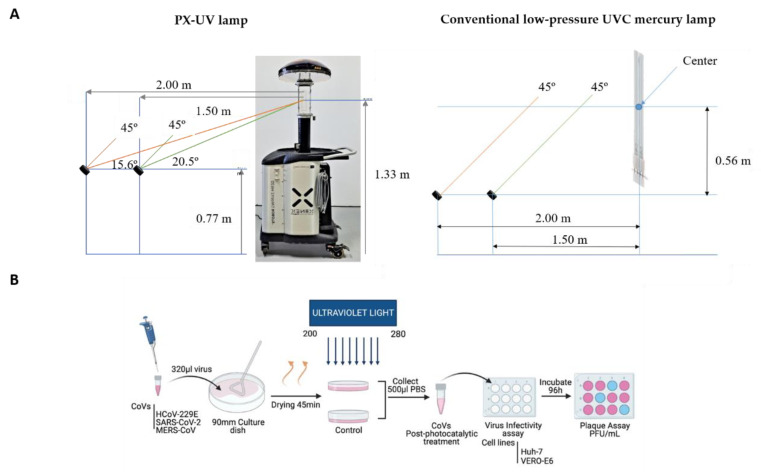
Schematic representation of the experimental conditions. (**A**) The UV light doses that samples received were measured by a detector anchored with an inclination of 45° and distances of 1.5 and 2 m with respect to the emission source. Distances were measured using a telemeter and the slope using a clinometer. (**B**) A 320 μL drop of DMEM, containing 10^5^ pfu of MERS-CoV, SARS-CoV-2, or HCoV-229E, was deposited on a 90 mm culture plate, spread, and dried for 45 min. After drying, viruses were irradiated with PX-UV for different time periods (0.5, 1, 2, 3, 4, and 5 min) at a 1.5 m or 2 m distance. Viruses were then collected in 500 µL of PBS to determine the infectivity by a plaque assay in Huh-7 (HCoV-229E and MERS-CoV) or Vero E6 (SARS-CoV-2) cell lines. This illustration was created with Biorender.com.

**Figure 2 ijerph-19-13780-f002:**
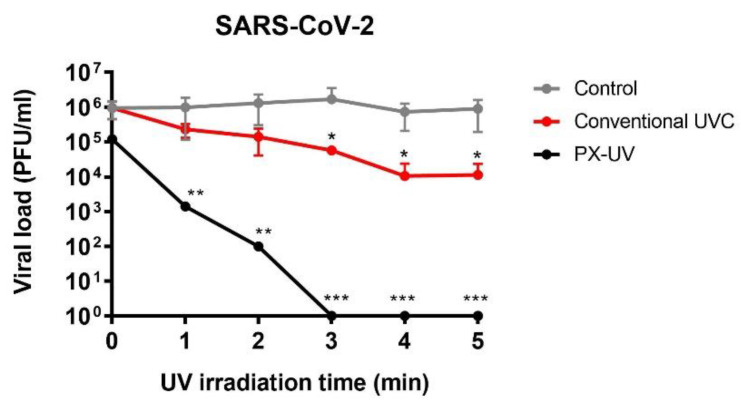
Comparison between the efficacy of conventional mercury UVC light and PX-UVC light in the decontamination of SARS-CoV-2 on plastic surfaces. A drop of 320 μL of SARS-CoV-2 was deposited and spread on a cell culture plate, as described in Figure 2. The virus was exposed to PX-UV light or conventional mercury UVC light for the indicated time at a 1.5 m or 2 m distance, respectively. Non-UV-treated controls were included in this experiment. Viruses collected from controls or after exposure to irradiation were titered. Viral load is represented as the mean of biological duplicates. Error bars indicate s.d. Viral titers of mock-irradiated samples remained constant in time. Statistical differences between treated and control groups are indicated: * *p* < 0.05; ** *p* < 0.01; *** *p* < 0.001).

**Table 1 ijerph-19-13780-t001:** Efficacy of PX-UV light in reducing human coronavirus titers on plastic surfaces at a 1.5 m distance.

		PX-UV-C Irradiation Time (min)						
		Control	0.5	1	2	3	4	5
HCoV-229E	Mean titer (PFU/mL)	2.1 × 10^5^ ± 2.1 × 10^4^	np	6.5 × 10^2^ ± 71	9.5 × 10 ± 49	4.5 × 10 ± 71	nd	nd
	Log reduction		np	2.5	3.4	3.7	>4.0	>4.0
SARS-CoV-2	Mean titer (PFU/mL)	1.2 × 10^5^ ± 4.9 × 10^3^	4.0 × 10^4^ ± 0.0	1.4 × 10^3^ ± 13	1.0 × 10^2^ ± 0.0	nd	nd	nd
	Log reduction		0.5	0.9	3.1	>3.8	>3.8	>3.8
MERS-CoV	Mean titer (PFU/mL)	2.2 × 10^5^ ± 1.1 × 10^4^	8.0 × 10^2^ ± 28	6.0 × 10 ± 57	nd	nd	nd	nd
	Log reduction		2.4	3.6	>4.0	>4.0	>4.0	>4.0

PFU, plaque-forming units; nd, non-detected; np, non-performed.

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
