# Peer review of "Pulsed-Xenon Ultraviolet Light Highly Inactivates Human Coronaviruses on Solid Surfaces, Particularly SARS-CoV-2"

_ijerph, 2022, doi:10.3390/ijerph192113780_

Round 1

Reviewer 1 Report

In the manuscript entitled “Pulsed-Xenon Ultraviolet Light Highly Inactivates Human 2 Coronaviruses on Solid Surfaces, ParticularlySARS-CoV-2”- the authors provide a brief report on the inactivation of human Coronaviruses using PX-UV irradiations. The experimental design is nicely organized, and this study is helpful for sanitization in real-life settings. Please find the following comments to improve the manuscript.

1.     Line 48-50, The spectral range of the UV light, with wavelengths between ~10 and 400 nm, corresponds to the electromagnetic radiation frame with the most energized photons that interact with matter by mostly electron excitation, rather than ionization – Please add the reference.

2.     Line 51, it would be better to mention the name of four bands and their range.

3.     Line 59-61, Among them, UVC is the principal UV germicidal irradiation (UVGI) because of its direct antimicrobial properties, which underlying mechanism of action is based on its efficient absorption by unsaturated organic compounds and the consequently disarrangement of their conjugated bonds that particularly and critically affect nucleic acids – Add the references after this line instead of clustering the references in line 63.

4. Line 61, change the word "consequently" to "consequent"

5.     Line 73-74, Novel devices implementing pulsed-light technologies offer improved capabilities in comparison to conventional steady-state lamps emitting restricted UV wavelengths- this sentence is little bit confusing. Please elaborate or rearrange.

6.     Methods and materials section, Line 113, generally for preparation of cell culture medium, double antibiotic (penicillin G and streptomycin) is needed. But, in this section, this part is missing. Please clarify.

7.     Line 237-239, CoV entry into cells depends on the specific cleavage of S glycoprotein to enhance fusion of viral and host cell membranes. CoV S protein contains at least two cleavage sites, S1/S2 and S2 – Add the refence.

Reviewer 2 Report

Line 99, please also add the time period.

Line 99, please add the method used to collect the swab

Please, consider adding a graphical abstract to increase interest in the manuscript.

Please, add information regarding ethical approval.

Reviewer 3 Report

The paper is very nicely written, the research has been conducted precisely, and the technique is thoroughly documented in the manuscript. I would recommend this research paper for publication since it is a good piece of work with high-quality research. However, there are certain issues with the manuscript that must be resolved before it can be accepted.
